# Predictors of Significant High-Sensitivity C-Reactive Protein Reduction After Use of Rosuvastatin/Amlodipine and Atorvastatin/Amlodipine—Subgroup Analysis in Randomized Controlled Trials [note 1]

**DOI:** 10.3390/jcm14103363

**Published:** 2025-05-12

**Authors:** Chun Muk Park, Hae Won Jung

**Affiliations:** Department of Cardiology, Daegu Catholic Medical Center, 33 Duryugongwonro 17-gil, Nam-gu, Daegu 42472, Republic of Korea; parkchm21@naver.com

**Keywords:** amlodipine, C-reactive protein, obesity, hydroxymethylglutaryl-CoA reductase inhibitors

## Abstract

**Introduction:** There are no clear predictors of high-sensitivity C-reactive protein (hsCRP) reductions following the use of antihypertensives and statins. Also, there are no clear data on the effect of BMI on hsCRP changes following the use of antihypertensives and statins. Therefore, we sought to identify predictors of significant hsCRP reduction after the use of rosuvastatin (RSV)/amlodipine (AML) and atorvastatin (ATV)/AML. **Methods:** We included 237 patients from 21 institutions in the Republic of Korea. Patients were randomly assigned to one of three treatment groups: RSV 10 mg/AML 5 mg; RSV 20 mg/AML 5 mg; or ATV 20 mg/AML 5 mg. Multivariate logistic regression analysis was performed to evaluate the predictors for hsCRP responders (hsCRP reduction ≥ 40% after 8 weeks). We also compared baseline hsCRP levels and their changes after 8 weeks between obese patients (*n* = 153) and nonobese patients (*n* = 84). **Results:** Baseline hsCRP ≥ 2 mg/dL and RSV 20 mg/AML 5 mg were independent predictors of hsCRP responders. Their median hsCRP % change rates were −53.11% and −40.0%, respectively. Normal weight, pre-obesity, and obesity were not independent predictors of hsCRP responders. The median hsCRP % reduction rates among normal weight, pre-obese, and obese patients were less than 40% in all groups, and the differences between each group were not significant (−20.0% vs. −33.33 vs. −23.08%, *p* = 0.289). **Conclusions:** In patients with ATV, RSV/AML polypill, baseline hsCRP ≥ 2 mg/dL, and baseline RSV 20 mg/AML 5 mg were independent predictors of a significant hsCRP reduction. BMI was not associated with hsCRP reduction (Clinical trial: NCT03951207).

## 1. Introduction

Regardless of low-density lipoprotein cholesterol (LDL-C) levels, C-reactive protein (CRP) has been well studied as an independent predictor of future cardiovascular events [1,2]. In addition, the role of statins in lowering CRP and high-sensitivity C-reactive protein (hsCPR) has also been well evaluated [3,4,5]. However, the target hsCRP level after the use of statins, especially in Asian patients, has not been determined. However, Kim-Mitsuyama et al. reported that a good clinical prognosis is shown if hsCRP decreases by more than 40% compared to before treatment in Japanese hypertensive patients [6]. And there are also no clear predictors of significant hsCRP reductions following antihypertensive medication and statin use. Obesity is one of the risk factors for elevated hsCRP, and a strong correlation between hsCRP and BMI has been shown in many studies [7,8,9]. However, there are no clear data on the effect of BMI on hsCRP before and after the use of antihypertensive medication and statins. Therefore, this study sought to identify clinical predictors, including BMI, of a significant reduction in hsCRP after the use of rosuvastatin (RSV)/amlodipine (AML) polypills and atorvastatin (ATV)/AML polypills.

## 2. Materials and Methods

### 2.1. Study Design and Participants of Study

This study is a post hoc sub-analysis of a trial, namely a “Randomized, multicenter, parallel, open, phase 4 study to compare the efficacy and safety of RSV/AML polypill versus ATV/AML polypill in hypertension patient with dyslipidemia (Clinical trial: NCT03951207) [10]”. The study design and the main results of the study have been described in our previous reports [10]. This clinical trial was conducted at 21 institutions in the Republic of Korea from May 2019 to September 2021. The principal and sub-investigators of each hospital recruited outpatients who met the inclusion and exclusion criteria. Subjects who met the inclusion/exclusion criteria were instructed to make therapeutic lifestyle changes from the time of screening. Subjects were administered (run-in) AML 5 mg for 4 weeks (±4 days) prior to randomization. Subjects receiving dyslipidemia treatment including a statin had a wash-out period of 4 weeks (±4 days) before randomization. Subjects receiving β-blockers or RAS inhibitors as antihypertensives maintained their dose unchanged from 4 weeks (±4 days) prior to randomization until the end of the study. After a wash-out/run-in period, patients eligible for randomization were finally enrolled and randomly assigned to 1 of 3 treatment groups: RSV 10 mg/AML 5 mg; RSV 20 mg/AML 5 mg; or ATV 20 mg /AML 5 mg. Randomization was performed in a 1:1:1 ratio by using SAS version 9.4 (SAS Institute Inc, Cary, NC, USA). During the 8-week treatment period, the assigned medications were administered once a day at the same time (morning) if possible. All patients were asked to visit the institution 4 and 8 weeks after randomization to assess the efficacy and safety (Figure 1). In this study, subjects whose medication compliance was less than 80% were excluded. This study complied with the Declaration of Helsinki and the Good Clinical Practice Guidelines defined by the International Council for Harmonization. It was approved by the institutional review board of each participating institution. Written informed consent was obtained from all individual participants included in the study.

### 2.2. Study Population

This study was conducted in patients with hypertension and dyslipidemia over the age of 19 years. This study was conducted with patients who were eligible for dyslipidemia medication by meeting the LDL-C criteria according to the risk group classification (Table 1). The detailed inclusion and exclusion criteria of this study were described in a previous article [10].

### 2.3. Calculation of Sample Size and Data Set Analyzed

In our previous study, we estimated the power of the test based on the assumption of a difference between the test and control groups in percent change in LDL-C at 8 weeks. In terms of LDL-C percent changes, we planned the enrolment of 324 patients (108 per treatment group) by adhering to the following criteria: (1) level of significance, α = 0.05 (superiority) and α = 0.025 (non-inferiority); (2) power of test = 80% and Type II error (β) = 0.2; (3) standard deviation = 14%; and (4) loss to follow-up = 20%. We set the non-inferiority criterion for LDL-C percent changes at 6% for the primary efficacy endpoint, as in previous statin trials [10]. We screened 381 subjects and excluded 122 subjects for the following reasons: inappropriate inclusion/exclusion criteria, *n* = 93, and withdrawal of subject consent, *n* = 24 and Etc: *n* = 5 (Figure 2). Finally, a total of 259 subjects were randomly assigned to each group: 86 subjects in the RSV 10 mg/AML 5 mg group (test group 1), 87 subjects in the RSV 20 mg/AML 5 mg group (test group 2), and 86 subjects in the ATV 20 mg /AML 5 mg group (control group). Among the 259 subjects, 1 subject each from test group 1 and test group 2 was excluded from the safety set for having had ‘no administration of clinical trial drugs’, and a total of 257 patients were used for the safety evaluation analysis data. Within the safety set, one person in test group 2 was excluded for ‘missing efficacy evaluation after baseline’, and a total of 256 people were used in the full analysis set (FAS). Within the FAS, 9 cases of ‘dropout’ and 10 cases of ‘significant protocol violation’ were excluded, and a total of 237 subjects were used in the per-protocol set (PPS) (Figure 2).

### 2.4. Endpoint

The primary endpoint of the “Randomized, multicenter, parallel, open, phase 4 study to compare the efficacy and safety of RSV/AML polypill versus ATV/AML polypill in hypertension patient with dyslipidemia” was the non-inferiority of RSV 10 mg/AML 5 mg compared to ATV 20 mg/AML 5 mg in terms of the LDL-C % change rate and the superiority of RSV 20 mg/AML 5 mg compared to ATV 20 mg/AML 5 mg in terms of the LDL-C% change rate after 8 weeks of administration. We performed sub-analysis with the PPS (*n* = 237). We categorized patients into two groups according to whether the achieved reduction in hsCRP after 8 weeks was equal to or greater than 40% (hsCRP responder group, *n* = 86) or less than 40% (hsCRP non-responder group, *n* = 151). Multivariate logistic regression analysis was performed to evaluate the predictors for hsCRP responders. In another sub-analysis, we compared baseline hsCRP levels and their changes after 8 weeks between the obese group (*n* = 153) and the nonobese group (*n* = 84). Regarding the definition and classification of obesity, we followed the 2022 Korean Obesity Guidelines for the definition of obesity (underweight: BMI < 18.5 kg/m^2^; normal weight: 18.5 kg/m^2^ ≤ BMI < 23 kg/m^2^; pre-obesity: 23 kg/m^2^ ≤ BMI < 25 kg/m^2^; obesity: 25 kg/m^2^ ≤ BMI) [11].

### 2.5. Statistical Analysis

Data were expressed as a number (%) and the mean ± standard deviation. A median test was performed to compare median hsCRP % changes. Categorical data were compared using the Chi-square test or Fisher’s exact test. Continuous variables were compared using Student’s *t*-test and the Kruskal–Wallis H test when they were normally and non-normally distributed, respectively. Univariate analysis using logistic regression was performed to identify potential independent predictors of hsCRP responders. Variables with a *p*-value< 0.05 in the univariate analysis were included in the multivariate analysis to identify independent predictors of hsCRP responders. A *p*-value < 0.05 was considered statistically significant. Statistical analyses were performed using SPSS version 25.0 (IBM, Armonk, NY, USA).

## 3. Results

The average age of the 237 patients was 62.73 ± 10.35 years, and male patients accounted for 65.40% of all the patients. The mean hsCRP% change rate at week 8 was 33.99 ± 337.43%. The median hsCRP% change rate at week 8 was −23.07% (IQR (interquartile −50–18.33%). The mean LDL-C change rate at week 8 was −47.94 ± 15.60%. The mean hsCRP change rates at week 8 for RSV 10 mg/AML 5 mg, RSV 20 mg/AML 5 mg, and ATV 20 mg/AML 5 mg were 13.4 ± 130.24%, −5.61 ± 178.27%, and 94.6 ± 538.14%, respectively. The median hsCRP% change rates at week 8 for RSV 10 mg/AML 5 mg, RSV 20 mg/AML 5 mg, and ATV 20 mg/AML 5 mg were −11.11% (IQR −46.41–25.0%), −40% (IQR −65.16–−7.14%), and −17.78% (IQR −37.50–25.0%), respectively.

### 3.1. Comparison Between hsCRP Responders and Non-Responders

We compared the hsCRP responder group (*n* = 86) and hsCRP non-responder group (*n* = 151) (Table 2). The proportions of patients with FRS > 20% and patients with RSV 20 mg/AML 5 mg were significantly higher in the responder group than in the non-responder group. The responder group showed a significantly higher FRS compared to the non-responder group (10.78 ± 7.01% vs. 12.88 ± 8.42%. *p* = 0.041). Baseline hsCRP, hsCRP % reduction on week 8, and LDL-C % reduction on week 8 were significantly higher in the responder group (baseline hsCRP: 0.89 ± 0.87 mg/dL vs. 2.08 ± 2.35 mg/dL, *p* < 0.0001; hsCRP% change on week 8: 88.37 ± 413.28% vs. −61.51 ± 15.06%, *p* = 0.001; LDL-C% change on week 8: −45.56 ± 16.49% vs. −52.12 ± 12.96%, *p* = 0.002). hsCRP on week 8 was significantly lower in the responder group (1.58 ± 3.69 mg/dL vs. 0.59 ± 0.51 mg/dL, *p* = 0.015). LDL-C on week 8, baseline total cholesterol (TC), TC on week 8, and apolipoprotein B (Apo B) on week 8 were significantly lower in the responder group.

### 3.2. Predictors of hsCRP Responders

According to the multivariate regression analysis, RSV 20 mg/AML 5 mg (odds ratio [OR]: 2.873, 95% confidence interval [CI]: 1.592–5.185, *p* < 0.0001) and baseline hsCRP ≥ 2 mg/dL (OR: 4.291, 95% CI: 2.108–8.738, *p* < 0.0001) were independently associated with hsCRP responders. Normal weight, pre-obesity, and obesity were not independent predictors of hsCRP responders (Table 3).

### 3.3. Comparison Between Baseline hsCRP < 2 mg/dL and Baseline hsCRP ≥ 2 mg/dL

We compared the baseline hsCRP < 2 mg/dL group (*n* = 191) and the baseline hsCRP ≥ 2 mg/dL group (*n* = 46) (Table 4). The baseline hsCRP ≥ 2 mg/dL group showed a significantly higher FRS, baseline hsCRP, and hsCRP at 8 weeks. In terms of hsCRP % changes at 8 weeks, there were no significant differences between the two groups (43.47 ± 358.56% vs. −5.42 ± 228.74, *p* = 0.379). The proportion of patients who were hsCRP responders was significantly higher in the baseline hsCRP ≥ 2 mg/dL group than in the baseline hsCRP < 2 mg/dL group (65.2% vs. 29.3%, *p* < 0.0001).

### 3.4. Comparison Between ATV 10 mg/AML 5 mg, RSV 10 mg/AML 5 mg, and RSV 20 mg/AML 5 mg

We compared the ATV 10 mg/AML 5 mg group, the RSV 10 mg/AML 5 mg group (*n* = 156), and the RSV 20 mg/AML 5 mg group (*n* = 81) (Table 4). There were no significant differences between the two groups in terms of baseline hsCRP, hsCRP at 8 weeks, and hsCRP % changes at 8 weeks. The RSV 20 mg/AML 5 mg group showed significantly lower Apo B levels at 8 weeks and LDL-C levels at 8 weeks and a greater LDL-C% reduction at 8 weeks. The proportion of patients who were hsCRP responders was significantly higher in the RSV 20 mg/AML group than in the ATV 10 mg/AML 5 mg group and the RSV 10 mg/AML 5 mg group (51.9% vs. 28.2%, *p* < 0.0001).

### 3.5. Comparison of hsCRP Difference and hsCRP % Change According to hsCRP Level and Treatment Group

Among patients with hsCRP < 1 mg/dL, 1 mg/dL ≤ hsCRP <2 mg/dL, and 2 mg/dL ≤ hsCRP, the mean hsCRP differences (baseline hsCRP-hsCRP at week 8) and the median hsCRP % reduction were significantly higher in the 2 mg/dL ≤ hsCRP group compared to the other two groups (Figure 3A,B). The median hsCRP % change in the 2 mg/dL ≤ hsCRP group was −53.12% (IQR −75.82–−15.00%), the only group to show a decrease of more than 40% among the three hsCRP groups (Figure 3B). Among patients with ATV 10 mg/AML 5 mg, RSV 10 mg/AML 5 mg, and RSV 20 mg/AML 5 mg, the mean hsCRP differences (baseline hsCRP-hsCRP at week 8) and the median hsCRP % reduction were significantly higher in the RSV 20 mg/AML 5 mg group compared to the other two treatment groups (Figure 3C,D). The median hsCRP% change in the RSV 20 mg/AML 5 mg group was −40.0% (IQR −65.16–−7.145). The other two groups showed a median hsCRP % reduction of less than 40% (Figure 3D).

### 3.6. Mean hsCRP at Baseline and Week 8 According to BMI and Mean hsCRP Differences and Median hsCRP % Change Between Baseline and Week 8 According to BMI

In normal-weight patients, pre-obese patients, and obese patients, the mean hsCRP at baseline and week 8 tended to increase as BMI increased (Figure 4A,B). There were no significant differences in the mean hsCRP differences and median hsCRP % change between baseline and week 8 according to BMI (Figure 4C,D). The median hsCRP % change did not exceed 40% in any of the three groups (Figure 4D).

### 3.7. Correlation Between Baseline hsCRP and hsCRP Difference and Correlation Between Baseline BMI and hsCRP Difference

In the correlation analysis between BMI and hsCRP difference, the correlation was weak and was not significant for r = 0.02 (*p* = 0.789) (Figure 5A). In the correlation analysis between baseline hsCRP and hsCRP difference, there was a strong negative correlation between baseline hsCRP and hsCRP differences for r = −0.385 (*p* < 0.0001). The formula for the simple linear regression was Y = −0.734X + 0.8760 (Figure 5B). In the ATV 20 mg/AML 5 mg group, the correlation between baseline hsCRP and hsCRP difference was weak and was not significant for r = −0.03 (*p* = 0.772) (Figure 5C). In the RSV 10 mg/AML 5 mg group and the RSV 20 mg/AML 5 mg group, the correlation between baseline hsCRP and hsCRP difference was strongly negative and significant (RSV 10 mg/AML 5 mg: r = −0.813, *p* < 0.0001; RSV 20 mg/AML 5 mg: r = −0.770, *p* < 0.0001) (Figure 5D,E).

## 4. Discussion

The primary findings of the present study are as follows: (1) The hsCRP % change after the use of ATV and RSV/AML polypills was heterogeneous. (2) Baseline hsCRP ≥ 2 mg/dL and RSV 20 mg/AML 5 mg were independent predictors of hsCRP responders. (3) The baseline hsCRP and the hsCRP after 8 weeks of taking ATV and RSV/AML polypills tended to increase with increasing BMI. (4) Normal weight, pre-obesity, and obesity were not associated with a significant hsCRP % reduction, and the hsCRP % reductions in each group were not significantly different. (4) In the correlation analysis between the baseline hsCRP and hsCRP difference, there was a strong negative correlation, while there was no significant correlation between BMI and hsCRP differences.

Elevated CRP and hsCRP are associated with worse cardiovascular outcomes, independently of LDL-C level [1,2]. Emerging evidence suggests that hsCRP may not only be a useful marker of inflammation but may also play an active role in the pathogenesis of atherosclerosis [12]. Therefore, determining the target level of hsCRP is an important issue. The JUPITER trial demonstrated that rosuvastatin treatment significantly reduced MACE in patients without hyperlipidemia but with elevated hsCRP levels. The results were based on a 50% reduction in LDL-C and a 37% reduction in hsCRP in the rosuvastatin treatment group [5]. However, the ASCOT trial failed to show that a reduction in CRP is related to cardiovascular (CV) outcomes. In the ASCOT trial, atorvastatin 10 mg reduced the median LDL-C by 40.3% and the median CRP by 27.4% [13]. Few studies have investigated the association between hsCRP reduction and CV outcomes in Asian patients. However, Kim-Mitsuyama et al. performed their studies with Japanese patients and reported that a good clinical prognosis is shown if hsCRP decreases by more than 40% compared to before antihypertensive medication [6]. Based on the above studies, we defined hsCRP responders as patients with a 40% or greater decrease in hsCRP. However, the clinical predictors of a more than 40% reduction in hsCRP after statin treatment have not been presented previously. Therefore, we conducted this study to identify clinical predictors of a more than 40% reduction in hsCRP after the administration RSV and ATV/AML polypills.

The role of statins in lowering CRP and hsCPR has been presented in many previous studies [3,4,5]. Xie et al. reported that statin use resulted in more pronounced reductions in hsCRP in a pooled analysis of trials with higher CRP levels at baseline [14]. Horiuchi et al. also demonstrated that statin therapy reduces inflammatory markers in dyslipidemia patients and that this anti-inflammatory effect is limited to patients who have high inflammatory markers at baseline (hsCRP ≥ 2.0 mg/dL) [15]. The results of our study are similar to those of previous studies. Baseline hsCRP ≥ 2 mg/dL was an independent predictor of hsCRP responders. The present study also revealed that the baseline hsCRP was significantly higher in the responder group, and the mean hsCPR of the responder group was over 2 mg/dL (baseline hsCRP: 0.89 ± 0.87 mg/dL vs. 2.08 ± 2.35 mg/dL, *p* < 0.0001). However, the baseline mean hsCRP in the obese group was 1.44 ± 1.83 mg/dL, which is less than 2 mg/dL.

Obesity is one of the risk factors for elevated hsCRP. A strong correlation between hsCRP and obesity has been shown in many studies [7,8,9]. However, there are no clear data on the effect of BMI on hsCRP before and after the use of antihypertensive medication and statins. Our study found that both baseline hsCRP and hsCRP on week 8 tended to increase with increasing BMI, and BMI was not associated with changes in hsCRP before and after the use of ATV and RSV/AML polypills. In the present study, before and after the use of ATV and RSV/AML polypills, the mean baseline hsCRP was highest in obese patients and the hsCRP difference was positive only in obese patients, which means that in obese patients, ATV and RSV/AML polypills were not effective at lowering hsCRP. In previous studies, the residual cardiovascular risk of obesity after statin use has been presented [16,17]. One of the reasons for this residual cardiovascular risk is inflammation in obese patients [16]. In a population with hyperlipidemia, Sandfort et al. showed that obese patients showed atheroma progression despite undergoing optimized statin therapy. In their study, the CRP levels in obese participants were significantly higher compared with nonobese participants, and CRP was associated with plaque progression [17].

In previous studies, authors have not find an association between LDL-C reduction and changes in CRP levels after statin use [14,15]. These articles suggests that hsCRP response after statin use is heterogeneous across patients. In an important previous study, the efficacy of ATV and RSV/AML in lowering LDL-C and blood pressure was consistent in most patients [10]. However, in the current study, the standard deviation % change in hsCRP was large at 337.43%, indicating heterogeneous responses in hsCRP among individuals taking ATV and RSV/AML polypills. Both the type and intensity of statins can affect the reduction in hsCRP after statin use [18]. Ma et al. reported that RSV produces a better reduction in CRP concentrations than ATV at a dose ratio of 1/1 and 1/2 (RSV/ATV), respectively [19]. Also, in our previous study, we reported that RSV 10 mg/AML 5 mg more effectively reduced LDL-C compared with ATV 20 mg /AML 5 mg [10]. However, in the present study, there were no significant differences in the median hsCRP % reduction between ATV 20 mg/AML 5 mg and RSV 10 mg/AML 5 mg (−17.78% vs. −11.1%, *p* = 0.523). Further studies comparing the hsCRP-reducing effects of RSV and ATV with the same intensities in larger patient populations are needed. To our knowledge, this is the first study to compare the hsCRP-reducing ability of ATV 20 mg/AML 5 mg, RSV 10 mg/AML 5 mg, and RSV 20 mg/AML 5 mg polypills. In this study, RSV 20 mg/AML 5 mg was confirmed to be an independent predictor of hsCRP responders and showed the highest median hsCRP % reduction among the three treatment groups, suggesting that high-intensity statins are more effective than moderate-intensity statins in reducing hsCRP, which is consistent with previous studies [20,21]. Zamani et al. also demonstrated that ATV 40 mg (40%) reduced hsCRP levels more than ATV 20 mg (13.3%) [22]. Taking these points into account, the reduction in hsCRP appears to depend more on the intensity than on the type of statin.

This study had several limitations. First, a relatively small number of patients were evaluated over a short period of time. Second, the mean BMI of obesity patient was 28.28 ± 2.92 Kg/m^2^ and the mean hsCRP at baseline of obesity patient was 1.44 ± 1.83 mg/dL. Considering that Westerners generally have more severe obesity and higher hsCRP than Asians, it seems difficult to apply or generalize the results of this study to other races. Third, there could be intraindividual CRP variation in our study. In the MESA cohort, Degoma et al. reported that the intraindividual variation in CRP was significantly greater than that for cholesterol measures [23]. Elevated CRP or hsCRP levels can be caused by a variety of factors and CRP changes can occur very quickly [24]. In the present study, at the time of the blood test, fever or infection symptoms were not checked. Also, chronic inflammation or malignancy of the patient was not considered when evaluating hsCRP. Therefore, hsCRP changes due to infection, chronic inflammation, or malignancy cannot be ruled out in this study. However, the present study has several strengths. This study first evaluated predictors of significant hsCRP reduction after using ATV and RSV/AML polypills. Second, this study is the first to examine the effect of obesity on hsCRP before and after using ATV and RSV/AML polypills. Third, we found a significant negative correlation between baseline hsCRP and hsCRP differences. In terms of hsCRP % reduction, predicting hsCRP changes after statin use can be difficult because of the heterogenicity. However, predicting hsCRP differences (absolute values) can be possible using baseline hsCRP, especially in patients administered RSV/AML polypills.

## 5. Conclusions

In patients with ATV and RSV/AML polypills, baseline hsCRP ≥ 2 mg/dL and RSV 20 mg/AML 5 mg were independent predictors of a significant hsCRP reduction. BMI was not associated with hsCRP reduction.

## Figures and Tables

**Figure 1 jcm-14-03363-f001:**
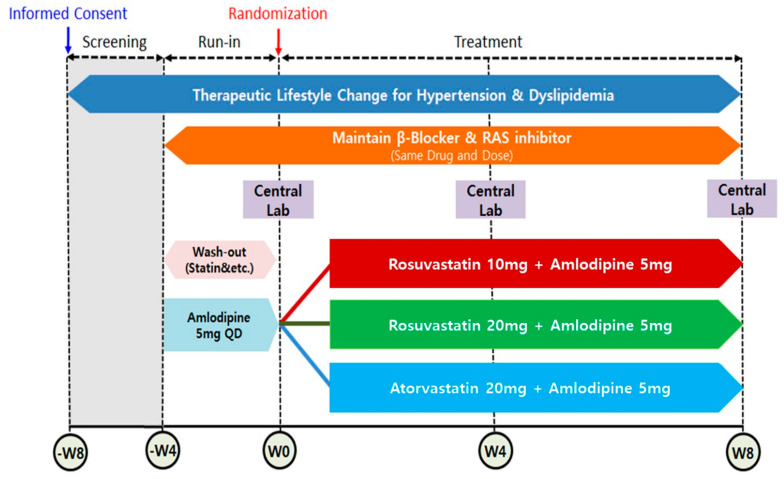
Schematic diagram of trial. RAS = renin angiotensin system.

**Figure 2 jcm-14-03363-f002:**
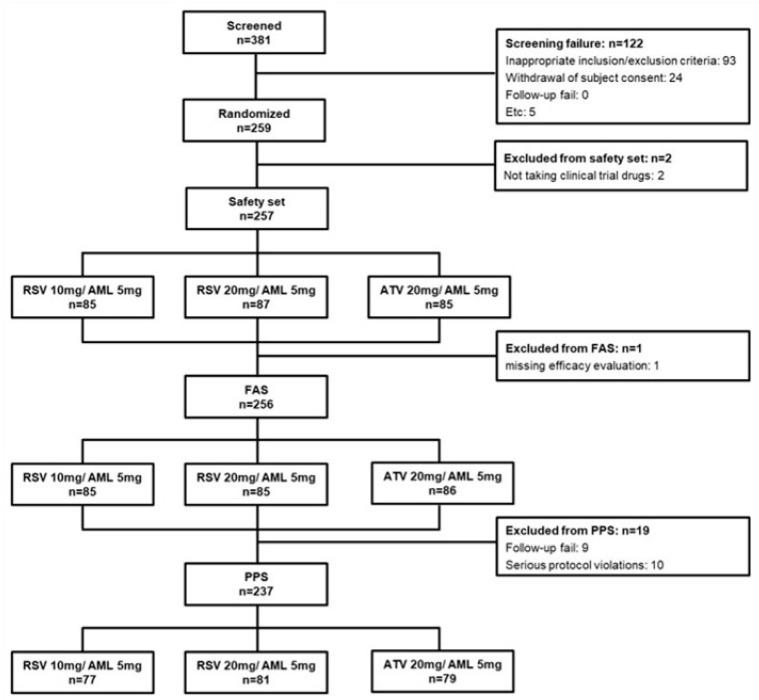
Enrollment flow chart for analysis. AML: amlodipine; ATV: atorvastatin; RSV: rosuvastatin; FAS: full analysis set; PPS: per-protocol set.

**Figure 3 jcm-14-03363-f003:**
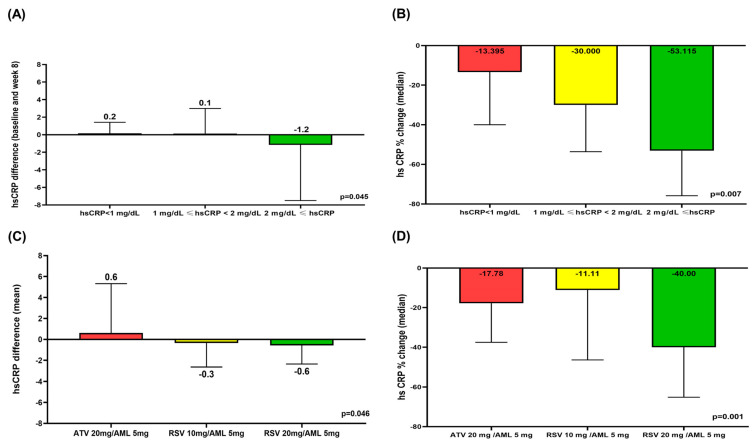
Mean hsCRP differences between baseline and week 8 according to hsCRP level (**A**). Median hsCRP% change between baseline and week 8 according to hsCRP level (**B**). Mean hsCRP differences between baseline and week 8 according to treatment group (**C**). Median hsCRP% change between baseline and week 8 according to treatment group (**D**). AML: amlodipine; ATV: atorvastatin; hsCRP: high-sensitivity C-reactive protein; RSV: rosuvastatin.

**Figure 4 jcm-14-03363-f004:**
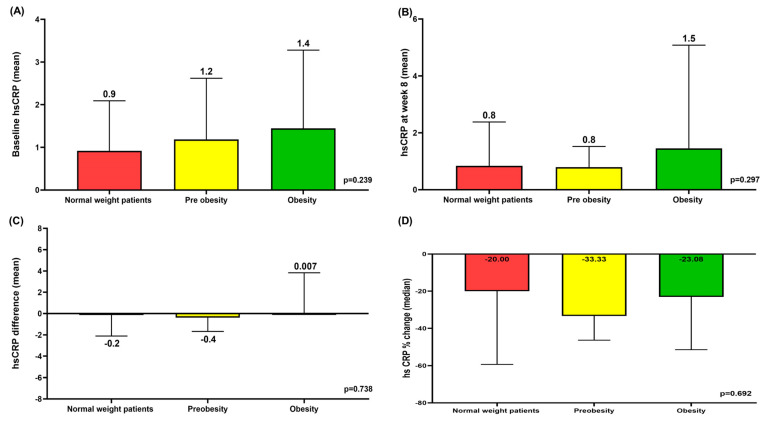
Mean baseline hsCRP according to BMI (**A**). Mean hsCRP on week 8 according to BMI (**B**). Mean hsCRP differences between baseline and week 8 according to BMI (**C**). Median hsCRP % change between baseline and week 8 according to BMI (**D**). BMI: body mass index; hsCRP: high-sensitivity C-reactive protein.

**Figure 5 jcm-14-03363-f005:**
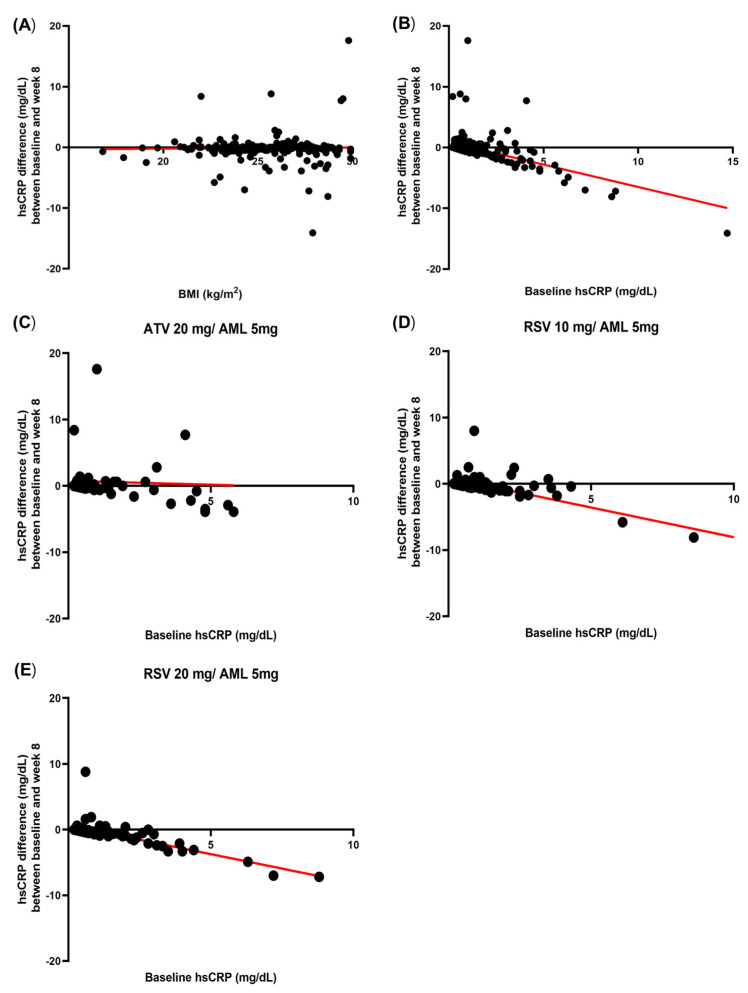
Correlation between baseline BMI and hsCRP difference (**A**). Correlation between baseline hsCRP and hsCRP difference (**B**). Correlation between baseline hsCRP and hsCRP difference in ATV 20 mg/AML 5 mg group (**C**). Correlation between baseline hsCRP and hsCRP difference in RSV 10 mg/AML 5 mg group (**D**). Correlation between baseline hsCRP and hsCRP difference in RSV 20 mg/AML 5 mg group (**E**). The dots represent each patient, and the red line represents the linear regression. AML: amlodipine; BMI: body mass index; ATV: atorvastatin; hsCRP: high-sensitivity C-reactive protein; RSV: rosuvastatin.

**Table 1 jcm-14-03363-t001:** Risk group classification.

Risk Group Classification		LDL-C (mg/dL)
Group I	No major cardiovascular risk factors other than hypertension or dyslipidemia	160–250
	- One or more cardiovascular risk factors other than hypertension and dyslipidemia - Framingham risk score < 10%	
Group II	- One or more cardiovascular risk factors other than hypertension and dyslipidemia - Framingham risk score 10~20%	130~250
Group III	- Coronary artery disease with hypertension and dyslipidemia or equivalent risk - Framingham risk score > 20%	100~250

Major cardiovascular risk factors: smoking, HDL-C < 40 mg/dL, age (male ≥ 45, female ≥ 55), and family history of early-onset coronary artery disease (among parents, siblings: male < 55, female < 65). If HDL-C ≥ 60 mg/dL, we subtracted one from the number of risk factors. HDL: high-density lipoprotein cholesterol; LDL-C: low-density lipoprotein cholesterol.

**Table 2 jcm-14-03363-t002:** Clinical characteristics of hsCRP responders and non-responders.

	Non-Responder (*n* = 151)	Responder (*n* = 86)	*p*-Value
Total patients, *n*	151	86	
Sex			0.618
Male, *n* (%)	97 (64.2)	58 (67.4)	
Female, *n* (%)	54 (35.8)	28 (32.6)	
Age, years	62.25 ± 10.22	63.56 ± 10.57	0.347
BMI, kg/m^2^	26.40 ± 3.56	26.46 ± 3.55	0.897
eGFR, mL/min/1.73 m^2^	89.32 ± 17.96	86.47 ± 18.99	0.251
Obesity, *n* (%)	95 (62.9)	58 (67.4)	0.483
Hypertension medication, *n* (%)	136 (90.1)	75 (87.2)	0.499
Duration of hypertension, years	9.49 ± 7.14	8.32 ± 6.85	0.222
Dyslipidemia medication, *n* (%)	88 (58.3)	55 (64.0)	0.390
Duration of dyslipidemia, years	6.28 ± 5.20	5.90 ± 5.76	0.606
Diabetes mellitus, *n* (%)	47 (31.1)	36 (41.9)	0.096
Smoking a month ago, *n* (%)	32 (21.2)	23 (26.7)	0.330
Drinking a month ago, *n* (%)	72 (47.7)	40 (46.5)	0.862
Myocardial infarction, *n* (%)	19 (12.6)	10 (11.6)	0.829
Angina, *n* (%)	36 (23.8)	28 (32.6)	0.146
Coronary revascularization, *n* (%)	23 (15.2)	11 (12.8)	0.606
PAD, AAA, symptomatic carotid disease, *n* (%)	9 (6.0)	6 (7.0)	0.757
FRS	10.78 ± 7.01	12.88 ± 8.42	0.041
FRS, *n* (%)			0.044
<10%, *n* (%)	69 (45.7)	31 (36.0)	
10~20%, *n* (%)	72 (47.7)	41 (47.7)	
>20%, *n* (%)	10 (6.6)	14 (16.3)	
Family history of premature CAD, *n* (%)	13 (8.6)	5 (5.8)	0.435
Categories of risk			0.419
Group I, *n* (%)	27 (17.9)	11 (12.8)	
Group II, *n* (%)	22 (14.6)	10 (11.6)	
Group III, *n* (%)	102 (67.5)	65 (75.6)	
Medication			0.001
RSV 10 mg/AML 5 mg, *n* (%)	60 (39.7)	19 (22.1)	
ATV 20 mg/AML 5 mg, *n* (%)	52 (34.4)	25 (29.1)	
RSV 20 mg/AML 5 mg, *n* (%)	39 (25.8)	42 (48.8)	
LDL-C, baseline, mg/dL	157.19 ± 29.47	149.33 ± 29.80	0.050
LDL-C, 8 weeks, mg/dL	84.45 ± 26.60	71.65 ± 23.69	<0.0001
LDL-C, 8 weeks, % changes	−45.56 ± 16.49	−52.12 ± 12.96	0.002
Total cholesterol, baseline, mg/dL	222.94 ± 32.12	213.45 ± 30.95	0.028
Total cholesterol, 8 weeks, mg/dL	149.91 ± 31.33	139.18 ± 25.69	0.007
HDL-C, baseline	46.70 ± 11.91	47.60 ± 11.15	0.567
HDL-C, 8 weeks	50.86 ± 12.34	51.94 ± 12.40	0.521
Triglyceride, baseline	192.31 ± 180.22	164.08 ± 78.12	0.169
Triglyceride, 8 weeks	129.14 ± 56.33	132.79 ± 52.58	0.624
Apo A-1, baseline	143.38 ± 23.35	143.58 ± 22.83	0.948
Apo A-1, 8 weeks	149.80 ± 25.45	155.01 ± 25.33	0.130
Apo B, baseline	135.59 ± 22.78	132.19 ± 22.33	0.268
Apo B, 8 weeks	83.03 ± 22.01	75.33 ± 19.43	0.007
Lipoprotein (a), baseline	38.71 ± 46.96	46.28 ± 67.86	0.313
Lipoprotein (a), 8 weeks % changes	43.58 ± 56.42	51.00 ± 79.29	0.404
hsCRP, baseline	0.89 ± 0.87	2.08 ± 2.35	<0.0001
hsCRP, 8 weeks	1.58 ± 3.69	0.59 ± 0.51	0.015
hsCRP, 8 weeks % changes	88.37 ± 413.28	−61.51 ± 15.06	0.001
FBG, baseline	114.78 ± 22.77	118.55 ± 25.18	0.239
FBG, 8 weeks	116.33 ± 24.04	119.52 ± 26.20	0.343
HbA1c, baseline	6.09 ± 0.69	6.20 ± 0.74	0.231
HbA1c, 8 weeks	6.11 ± 0.75	6.31 ± 0.98	0.083

AAA: abdominal aorta aneurysm; AML: amlodipine; Apo A1: apolipoprotein A1; Apo B: apolipoprotein; ATV: atorvastatin; BMI: body mass index; CAD: coronary artery disease; eGFR: estimated glomerular filtration rate; FBG: fasting blood glucose; FRS: Framingham risk score; HDL-C: high-density lipoprotein cholesterol; HbA1C: hemoglobin a1c; hsCRP: high-sensitivity C-reactive protein; LDL-C: low-density lipoprotein cholesterol; PAD: peripheral artery disease; RSV: rosuvastatin.

**Table 3 jcm-14-03363-t003:** Independent predictors of hsCRP responders.

Variance	Univariate Analysis			Multivariate Analysis		
	HR	95% CI	*p*	HR	95% CI	*p*
Age	1.013	0.987–1.039	0.346			
Male	1.153	0.658–2.020	0.618			
Diabetes mellitus	1.593	0.919–2.761	0.097			
Obesity	1.221	0.698–2.135	0.484			
Pre-obesity	0.817	0.426–1.567	0.542			
Normal body weight	0.908	0.415–1.986	0.809			
Baseline hsCRP ≥ 2 mg/dL	4.520	2.285–8.940	<0.0001	4.291	2.108–8.738	<0.0001
LDL-C at baseline	0.991	0.982–1.000	0.052			
HDL-C at baseline	1.007	0.984–1.030	0.565			
Systolic BP at baseline	0.988	0.977–1.019	0.843			
Previous hypertension medication	0.752	0.329–1.720	0.752			
RSV 20 mg/AML 5 mg	2.741	1.569–4.790	<0.0001	2.873	1.592–5.185	<0.0001
FRS > 20%	2.742	1.160–6.477	0.021	2.375	0.936–6.025	0.069
Smoking a month ago	1.358	0.733–2.515	0.331			
Drinking a month ago	0.954	0.561–1.622	0.862			
eGFR	0.991	0.977–1.006	0.251			
Family history of premature CAD	0.655	0.225–1.905	0.438			
History of Myocardial infarction	0.914	0.404–2.067	0.829			
PAD, AAA, symptomatic carotid disease	1.183	0.406–3.445	0.758			
Coronary revascularization	0.816	0.377–1.768	0.607			

AAA: abdominal aorta aneurysm; AML: amlodipine; BP: blood pressure; CAD: coronary artery disease; CI: confidence interval; eGFR: estimated glomerular filtration rate; FRS: Framingham risk score; HR: hazard ratio; HDL-C: high-density lipoprotein cholesterol; hsCRP: high-sensitivity C-reactive protein; LDL-C: low-density lipoprotein cholesterol; PAD: peripheral artery disease; RSV: rosuvastatin; TC: total cholesterol.

**Table 4 jcm-14-03363-t004:** Comparison between the baseline hsCRP < 2 mg/dL group and the baseline hsCRP ≥ 2 mg/dL group and comparison between the ATV 10 mg/AML 5 mg, RSV 10 mg/AML 5 mg, and RSV 20 mg/AML 5 mg groups.

	Baseline hsCRP < 2 mg/dL *n* = 191	Baseline hsCRP ≥ 2 mg/dL *n* = 46	*p*-Value	ATV 10 mg/AML 5 mg, RSV 10 mg/AML 5 mg *n* = 156	RSV 20 mg/AML 5 mg *n* = 81	*p*-Value
Sex			0.508			0.768
Male, *n* (%)	123 (64.4)	32 (69.6)		101 (64.7)	54 (66.7)	
Female, *n* (%)	68 (35.6)	14 (30.4)		55 (35.3)	27 (33.3)	
Age, years	62.99 ± 10.27	61.63 ± 10.70	0.423	62.77 ± 10.64	62.64 ± 9.81	0.925
BMI, kg/m^2^	26.20 ± 3.49	27.37 ± 3.66	0.044	26.43 ± 3.59	26.42 ± 3.48	0.987
eGFR, mL/min/1.73 m^2^	88.55 ± 17.78	87.15 ± 20.71	0.642	88.82 ± 19.27	87.25 ± 16.51	0.536
Obesity (BMI ≥ 25 kg/m^2^)	123 (64.4)	32 (69.6)	0.508	104 (66.7)	49 (60.5)	0.346
Hypertension medication, *n* (%)	172 (90.1)	39 (84.8)	0.305	139 (89.1)	72 (88.9)	0.960
Duration of hypertension, years	9.46 ± 6.87	7.43 ± 7.59	0.080	9.38 ± 7.13	8.45 ± 6.87	0.337
Dyslipidemia medication, *n* (%)	118 (61.8)	25 (54.3)	0.355	91 (58.3)	52 (64.2)	0.381
Duration of dyslipidemia, years	6.47 ± 5.27	4.78 ± 5.79	0.056	6.33 ± 5.23	5.77 ± 5.72	0.449
Diabetes mellitus, *n* (%)	68 (35.6)	15 (32.6)	0.702	60 (38.5)	23 (28.4)	0.123
Smoking a month ago, *n* (%)	41 (21.5)	14 (30.4)	0.196	37 (23.7)	18 (22.2)	0.796
Drinking a month ago, *n* (%)	88 (46.1)	24 (52.2)	0.457	76 (48.7)	36 (44.4)	0.532
Myocardial infarction, *n* (%)	27 (14.1)	2 (4.3)	0.069	20 (12.8)	9 (11.1)	0.703
Angina, *n* (%)	53 (27.7)	11 (23.9)	0.599	41 (26.3)	23 (28.4)	0.728
Coronary revascularization, *n* (%)	31 (16.2)	3 (6.5)	0.092	23 (14.7)	11 (13.6)	0.809
PAD, AAA, symptomatic carotid disease, *n* (%)	13 (6.8)	2 (4.3)	0.539	7 (4.5)	8 (9.9)	0.106
FRS, %	11.03 ± 7.25	13.67 ± 8.68	0.034	11.53 ± 7.86	11.58 ± 7.12	0.958
FRS			0.055			0.802
<10%, *n* (%)	84 (44.0)	16 (34.8)		68 (43.6)	32 (39.5)	
10~20%, *n* (%)	92 (48.2)	21 (45.7)		72 (46.2)	41 (50.6)	
>20%, *n* (%)	15 (7.9)	9 (19.6)		16 (10.3)	8 (9.9)	
Family history of premature CAD, *n* (%)	14 (7.3)	4 (8.7)	0.754			
Categories of risk			0.690			0.255
Group I, *n* (%)	29 (15.2)	9 (19.6)		25 (16.0)	13 (16.0)	
Group II, *n* (%)	27 (14.1)	5 (10.9)		17 (10.9)	15 (18.5)	
Group III, *n* (%)	135 (70.7)	32 (69.6)		114 (73.1)	53 (65.4)	
Medication			0.654			
AML/RSV 10, *n* (%)	66 (34.6)	13 (28.3)				
AML/ATV 20, *n* (%)	62 (32.5)	15 (32.6)				
AML/RSV 20 *n* (%)	63 (33.0)	18 (39.1)				
SBP at baseline, mmHg	131.80 ± 12.33	132.40 ± 14.67	0.773	132.17 ± 13.21	131.42 ± 11.99	0.669
SBP at 8 weeks, mmHg	131.29 ± 14.16	133.56 ± 13.43	0.326	132.60 ± 13.88	130.06 ± 14.23	0.186
LDL-C, baseline, mg/dL	154.34 ± 29.62	154.34 ± 30.73	1.000	153.42 ± 30.79	156.11 ± 27.79	0.512
LDL-C, 8 weeks, mg/dL	80.11 ± 27.21	78.54 ± 22.16	0.716	83.37 ± 26.28	72.93 ± 24.98	0.004
LDL-C, 8 weeks % changes	−47.82 ± 15.92	−48.44 ± 14.32	0.809	−45.32 ± 15.35	−52.99 ± 14.91	<0.0001
Total cholesterol, baseline, mg/dL	219.44 ± 32.21	219.71 ± 31.29	0.959	218.83 ± 32.65	220.77 ± 30.76	0.658
Total cholesterol, 8 weeks, mg/dL	145.69 ± 30.95	147.36 ± 24.71	0.733	148.57 ± 30.76	141.11 ± 27.39	0.068
HDL-C, baseline, mg/dL	46.96 ± 11.42	47.30 ± 12.58	0.859	46.64 ± 11.83	47.77 ± 11.26	0.477
HDL-C, 8 weeks, mg/dL	50.84 ± 11.62	52.95 ± 15.05	0.300	50.68 ± 11.91	52.35 ± 13.17	0.324
Triglyceride, baseline, mg/dL	184.06 ± 163.79	173.76 ± 86.07	0.680	189.88 ± 179.47	167.01 ± 71.94	0.272
Triglyceride, 8 weeks, mg/dL	129.76 ± 56.43	133.36 ± 48.59	0.691	131.17 ± 53.99	129.11 ± 56.97	0.785
Apo A-1, baseline, mg/dL	143.39 ± 22.73	143.69 ± 24.90	0.938	143.34 ± 23.50	143.66 ± 22.48	0.921
Apo A-1, 8 weeks, mg/dL	150.42 ± 24.10	156.97 ± 30.26	0.118	150.86 ± 25.46	153.30 ± 25.60	0.486
Apo B, baseline, mg/dL	134.00 ± 22.76	135.82 ± 22.78	0.626	133.55 ± 22.86	135.91 ± 22.25	0.449
Apo B, 8 weeks, mg/dL	80.17 ± 22.44	80.49 ± 16.56	0.928	82.42 ± 21.43	76.03 ± 20.82	0.029
Lipoprotein(a), baseline, mg/dL	39.91 ± 52.51	47.88 ± 66.50	0.383	37.99 ± 46.84	48.15 ± 68.88	0.182
Lipoprotein(a), 8 weeks, mg/dL	45.64 ± 62.81	48.91 ± 76.72	0.762	42.68 ± 57.20	53.20 ± 79.19	0.242
hsCRP, baseline, mg/dL	0.69 ± 0.41	3.94 ± 2.31	<0.0001	1.28 ± 1.74	1.40 ± 1.55	0.606
hsCRP, 8 weeks, mg/dL	0.84 ± 1.74	2.79 ± 5.58	<0.0001	1.42 ± 3.58	0.83 ± 1.15	0.149
hsCRP, 8 weeks % changes	43.47 ± 358.56	−5.42 ± 228.74	0.379	54.55 ± 394.60	−5.61 ± 178.27	0.194
hsCRP responder, *n* (%)	56 (29.3)	30 (65.2)	<0.0001	44 (28.2)	42 (51.9)	<0.0001
FBG, baseline, mg/dL	116.15 ± 24.30	116.15 ± 21.25	0.999	117.36 ± 21.62	113.85 ± 27.22	0.281
FBG, 8 weeks, mg/dL	116.72 ± 23.61	120.67 ± 29.48	0.334	118.87 ± 24.93	114.82 ± 24.59	0.235
HbA1c, baseline, %	6.10 ± 0.70	6.25 ± 0.72	0.213	6.18 ± 0.67	6.03 ± 0.76	0.129
HbA1c, 8 weeks, %	6.16 ± 0.84	6.30 ± 0.87	0.312	6.22 ± 0.75	6.11 ± 1.00	0.347

AAA: abdominal aorta aneurysm; AML: amlodipine; Apo A1: apolipoprotein A1; Apo B: apolipoprotein; ATV: atorvastatin; BMI: body mass index; CAD: coronary artery disease; eGFR: estimated glomerular filtration rate; FBG: fasting blood glucose; FRS: Framingham risk score; HDL-C: high-density lipoprotein cholesterol; HbA1C: hemoglobin a1c; hsCRP: high-sensitivity C-reactive protein; LDL-C: low-density lipoprotein cholesterol; PAD: peripheral artery disease; RSV: rosuvastatin; SBP: systolic blood pressure.

## Data Availability

Unavailable due to personal information.

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
