# Peer review of "Predictors of Significant High-Sensitivity C-Reactive Protein Reduction After Use of Rosuvastatin/Amlodipine and Atorvastatin/Amlodipine—Subgroup Analysis in Randomized Controlled Trials†"

_jcm, 2025, doi:10.3390/jcm14103363_

Round 1

Reviewer 1 Report

Comments and Suggestions for Authors

The study analyzed predictors of a significant reduction in high-sensitivity C-reactive protein (hsCRP) in patients with hypertension and dyslipidemia treated with polypills containing rosuvastatin/amlodipine or atorvastatin/amlodipine. The combination of RSV 20 mg/AML 5 mg and a Framingham Risk Score (FRS) >20% were identified as independent predictors of a favorable inflammatory response. Although obesity was associated with higher hsCRP levels, it did not significantly influence hsCRP reduction

  Strength 

  1. Pioneering analysis: This is the first study to identify clinical predictors of significant hsCRP reduction (>40%) following treatment with rosuvastatin/amlodipine or atorvastatin/amlodipine polypills.

  2. Real-world, multicenter approach: Conducted across 21 outpatient centers, the study reflects real-world clinical practice.

  3. Assessment of obesity's impact: It is the first to directly compare the effect of obesity on hsCRP levels before and after statin/amlodipine treatment.

  4. Clinical relevance: The study suggests using the Framingham Risk Score (FRS) not only to decide whether to initiate statin therapy but also to choose its intensity for optimal anti-inflammatory effects.

  5. Focus on an Asian population: Provides valuable data on a group often underrepresented in clinical trials, with implications for personalized medicine.

Limitations

  1. Small sample size and short follow-up: The study included only 237 patients and assessed outcomes at 8 weeks, limiting insights into long-term benefits.

  2. Mild obesity in the cohort: The mean BMI of obese patients was 28, which may not reflect the effects in populations with more severe obesity, such as Western populations.

  3. Potential confounding from inflammation: The study did not account for fever or infections at the time of blood collection, which could have influenced hsCRP levels.

  4. Limited generalizability: Results from this Asian population may not be fully applicable to other ethnic groups with different metabolic and inflammatory profiles.

  5. Post-hoc analysis: As a sub-analysis of an existing trial, the findings may lack the methodological strength of a study specifically designed to assess hsCRP reduction

This is a lot of data that probably comes from an approval study.

C-reactive protein is not yet a widely used marker in clinical practice and the Framingham score is no longer used by anyone, except in the research field

These are important numbers in any case

Author Response

Dear reviewer.

Thank you for reviewing my paper.

We noticed that in our previous submission, we did not input baseline hsCRP as a variable in logistic regression. As a result of entering baseline hsCRP into logistic regression, we found that baseline hsCRP ≥ 2 and RSV 20 mg/AML 5 mg were independent prognostic factors for hsCRP responder. FRS was not were independent prognostic factor for hsCRP responder. Afterwards, we conducted a major revision following the above contents. We believe that the new analyses are more consistent with the existing papers on hsCRP and can be a positive addition to existing medical knowledge. Please review it again.

From Doctor Jung.

Reviewer 2 Report

Comments and Suggestions for Authors

This manuscript addresses a clinically relevant question regarding predictors of hsCRP reduction following statin/amlodipine polypill therapy, with a focus on obesity and cardiovascular risk as quantified by Framingham Risk Score. However, the following concerns limit its current impact:

The selection of ≥40% reduction in hsCRP as the cutoff for response, although previously reported in a Japanese cohort, should be further justified with reference to broader literature or statistical reasoning. Were alternative cutoffs considered? Consider modeling delta hsCRP as a continuous variable in regression to explore dose-response relationships.

The enormous standard deviations in hsCRP change (±337%) suggest high variability. It would be helpful to address the presence of potential outliers or to consider using median (IQR) values or log-transformation to normalize the distribution and better reflect central tendency.

While baseline hsCRP differs significantly between responders and non-responders, it was not included in the multivariate model. Given its strong association, the omission should be explained or its potential multicollinearity with FRS or statin dose should be addressed.

The manuscript concludes that obesity is not associated with hsCRP reduction. However, the sample may be underpowered for this analysis, particularly since the obese group showed a non-significant numerical trend toward lesser reduction.

No assessment of infection, recent illness, or concurrent inflammatory conditions was performed. This limits the internal validity regarding changes in hsCRP. This limitation should be more explicitly acknowledged in the discussion.

The conclusion that RSV 20 mg is superior may oversimplify the role of statin intensity. It would strengthen the discussion to include broader literature supporting a class effect vs. molecule-specific differences.

Author Response

Dear reviewer.

Thank you for reviewing my paper.

We noticed that in our previous submission, we did not input baseline hsCRP as a variable in logistic regression. As a result of entering baseline hsCRP into logistic regression, we found that baseline hsCRP ≥ 2 mg/dL and RSV 20 mg/AML 5 mg were independent prognostic factors for hsCRP responder. FRS was not were independent prognostic factors for hsCRP responder. Afterwards, we conducted a major revision following the above contents. We believe that the new analysis is more consistent with the existing papers on hsCRP and can be a positive addition to existing medical knowledge. Please review it again. Following is the answer for your comments.

Thank you

From Doctor Jung

Comment 1: The selection of ≥40% reduction in hsCRP as the cutoff for response, although previously reported in a Japanese cohort, should be further justified with reference to broader literature or statistical reasoning. Were alternative cutoffs considered?

Response1: Thank you for your comment. We added the evidence of the selection of ≥40% reduction in hsCRP as the cutoff for response in the discussion part. This change can be found page 14

Comment 2: Consider modeling delta hsCRP as a continuous variable in regression to explore dose-response relationships.

Response2: Thank you for your comment. Following your advice, we performed correlation analysis between delta hsCRP and baseline hsCRP. This change can be found page 11, 13 (figure 5).

Comment 3: The enormous standard deviations in hsCRP change (±337%) suggest high variability. It would be helpful to address the presence of potential outliers or to consider using median (IQR) values or log-transformation to normalize the distribution and better reflect central tendency.

Response 3: Thank you for your comment. Following your advice, we used median value of hsCRP % change. We made new graphs for it. This change can be found page 11, 12 (figure 3, 4).

Comment 4: While baseline hsCRP differs significantly between responders and non-responders, it was not included in the multivariate model. Given its strong association, the omission should be explained or its potential multicollinearity with FRS or statin dose should be addressed.

Response 4: Thanks for pointing out this very important point. Due to our lack of thoughtfulness, we did not include baseline hsCRP as a variable in the logistic regression. As a result of entering baseline hsCRP ≥ 2 mg/dL into logistic regression, we found that baseline hsCRP ≥ 2 mg/dL and RSV 20 mg/AML 5 mg were independent prognostic factors for hsCRP responder. This change can be found page 7, 8 (Table 3).

Comment 5: The manuscript concludes that obesity is not associated with hsCRP reduction. However, the sample may be underpowered for this analysis, particularly since the obese group showed a non-significant numerical trend toward lesser reduction.

Response 5: Considering the small number of patients and the small number of severely obese patients in this study, it is difficult to generalize this study. However, this study showed that the median hsCRP % reduction did not significantly differ between BMI groups. Since hsCRP is generally higher in obese patients than in normal-weight patients, we aimed to show that hsCRP may remain high even after statin use if the response to statin is similar. This change can be found page 12 (figure 4).

Comment 6: No assessment of infection, recent illness, or concurrent inflammatory conditions was performed. This limits the internal validity regarding changes in hsCRP. This limitation should be more explicitly acknowledged in the discussion.

Response 6: Thank you for your comment. Following your advice, we have described the limitations of this study in more detail. This change can be found page 15.

Comment 7: The conclusion that RSV 20 mg is superior may oversimplify the role of statin intensity. It would strengthen the discussion to include broader literature supporting a class effect vs. molecule-specific differences

Response 7: Thank you for your comment. When considering the results of this study and previous studies, it seems that the decrease in hsCRP is more affected by the intensity of statin than by the type of statin. Therefore, comprehensive evidence and opinions on this were described in the discussion part. This change can be found page 14, 15.

Round 2

Reviewer 2 Report

Comments and Suggestions for Authors

I would like to thank the authors for addressing my concerns.